# Multi-winner Approval Voting Goes Epistemic

**Tahar Allouche**[1]        **Jérôme Lang**[1]        **Florian Yger**[1]

[1]LAMSADE, CNRS, PSL, Université Paris-Dauphine,

## Abstract

Epistemic voting interprets votes as noisy signals about a ground truth. We consider contexts where the truth consists of a set of objective winners, knowing a lower and upper bound on its cardinality. A prototypical problem for this setting is the aggregation of multi-label annotations with prior knowledge on the size of the ground truth. We posit noise models, for which we define rules that output a set of winners corresponding to local maxima of the data likelihood function. We report on experiments on multi-label annotations (which we collected).

## 1 INTRODUCTION

The epistemic view of voting assumes the existence of a ground truth which, usually, is either an alternative or a ranking over alternatives. Votes reflect opinions or beliefs about this ground truth; the goal is to aggregate these votes so as to identify it. Usual methods define a noise model specifying the probability of each voting profile given the ground truth, and output the alternative that is the most likely state of the world, or the ranking that is most likely the true ranking.

Now, there are contexts where the ground truth does not consist of a single alternative nor a ranking, but of a *set of alternatives*. Typical examples are multi-label crowdsourcing (find the items in a set that satisfy some property, *e.g.* the sport teams appearing on a picture) or finding the objectively $k$ best candidates (best papers at a conference, best performance in artistic sports, $k$ patients with highest probabilities of survival if being assigned a scarce medical resource).

These alternatives that are truly in the ground truth are called 'winning' alternatives. Depending on the context, the number of winning alternatives can be fixed, unconstrained, or

more generally, constrained to be in a given interval. This constraint expresses some *prior knowledge on the cardinality of the ground truth*. This prior knowledge is held by the central authority that aggregates the votes, and not necessarily by the voters themselves. Here are some examples:

- *Picture annotation via crowdsourcing*: participants are shown a picture taken from a soccer match and have to identify the team(s) appearing in it. The ground truth is known to contain one or two teams.
- *Guitar chord transcription*: voters are base classifier algorithms Nguyen et al. [2020] which, for a given chord, select the set of notes constitute it. The true set of notes can contain three to six alternatives.
- *Jury*: participants are members of a jury which has to give an award to three papers presented at a conference: the number of objective winners is fixed to three. (In a variant, the number of awards would be *at most* three.)

We assume that voters provide a simple form of information: *approval ballots*, indicating which alternatives they consider plausible winners. These approval ballots are not subject to any cardinality constraint: *a voter may approve a number of alternatives, even if it does not lie in the interval bearing on the output*. This is typically the case for totally ignorant voters, who may plausibly approve all alternatives.

Sometimes, the aggregating mechanism has some prior information about the likelihood of alternatives and the reliability of voters. We first study a simple case where this information is specified in the input: in the noise model, each voter has a probability $p_i$ (resp. $q_i$) of approving a winning (resp. non-winning) alternative, and each alternative has a prior probability to be winning. This departs from classical voting, where voters are usually treated equally (*anonymity*), and similarly for alternatives (*neutrality*).

This simple case serves as a building component for the more complex case where these parameters are not known beforehand but *estimated from the votes*: votes allow to infer information about plausibly winning alternatives, from which we infer information about voter reliabilities, which

*Accepted for the 38th Conference on Uncertainty in Artificial Intelligence* (UAI 2022).

leads to revise information about winning alternatives, and so on until the process converges to a local optimum of the likelihood function. Here we move back to an anonymous and neutral setting, since all alternatives (resp. voters) are treated equally before votes are known.

After discussing related work (Section 2), we introduce the model (Section 3) and give an estimation algorithm (Section 4), first in the case where the parameters are known, and then in the case where they are estimated from the votes. In Section 5 we present a data gathering task and analyse the results of the experiments. Section 6 concludes.

## 2 RELATED WORK

**Epistemic social choice** Epistemic social choice consists in recovering an objective *ground truth* from votes seen as noisy reports about the ground truth, using maximum likelihood estimation. It dates back from Condorcet's *jury theorem* [Condorcet, 1785]: $n$ independent, equally reliable voters vote on two alternatives that are *a priori* equally likely; if every vote is correct with probability $p > \frac{1}{2}$, then majority outputs the correct alternative with a probability increasing with $n$ and tending to 1 when $n$ grows to infinity.

There are several extensions of Condorcet's jury theorem: Young [1988] for an arbitrary number of alternatives; Shapley and Grofman [1984] and Drissi-Bakhkhat and Truchon [2004] for voters with various competence degrees; Ben-Yashar and Nitzan [1997] and Ben-Yashar and Paroush [2001] for nonuniform priors over alternatives; Pivato [2013] and Pivato [2017] for dependent voters. Conitzer and Sandholm [2005] and Conitzer et al. [2009] characterize various voting rules as maximum likelihood estimators, each associated with a particular noise model. See Nitzan and Paroush [2017] and Elkind and Slinko [2016]. for surveys on recent developments.

**Multi-winner voting** Multi-winner voting rules map voting profiles into sets of alternatives. A voting profile can be either a collection of subsets of alternatives (approval ballots) or a collection of ranking over alternatives (ordinal ballots). The output is often constrained to have a fixed cardinality, but not always: see Kilgour [2016], Faliszewski et al. [2020]. There have been a lot of recent developments in the field: see the recent surveys Faliszewski et al. [2017]) and Lackner and Skowron [2020]. They, however, deal only with the classical (non-epistemic) view of social choice, where votes express preferences.

**Multi-winner epistemic voting** Multi-winner epistemic voting has received only little attention so far. Procaccia et al. [2012] assume a ground truth ranking over alternatives, and identify rules that output the $k$ alternatives maximizing the likelihood to contain the best alternative, or the likelihood to coincide with the top-$k$ alternatives. The last section of

[Xia and Conitzer, 2011] defines a noise model where the ground truth is a set of $k$ alternatives (and the reported votes are partial orders). The only work we know where the noise models produce random *approval votes* from a ground truth consisting of *a set of alternatives* is [Caragiannis et al., 2020]. They define a family of distance-based noise models, whose prototypical instance generates approval votes selecting an alternative in the ground truth (resp. not in the ground truth) with probability $p$ (resp. $1 - p$); as we see further, this is a specific case of our noise model. Generalizing multiwinner voting, Xia et al. [2010] study epistemic voting on *combinatorial* (or *multi-attribute*) domains.

**Epistemic approval voting** Epistemic voting with approval ballots has scarcely been considered. Procaccia and Shah [2015] assume that the ground truth is a *ranking* over alternatives, and identify noise models for which approval voting is optimal given $k$-approval votes, in the sense that the objectively best alternative gets elected. Allouche et al. [2022] continue this line of research but assume instead that the ground truth consists of a single alternative. They define various noise models and show that those that work best on real datasets are those that give a higher confidence to voters who approve few alternatives. Caragiannis and Micha [2017] study the number of samples needed to recover the ground truth ranking over alternatives with high enough probability from approval ballots; they show that is is exponential if ballots are required to approve $k$ candidates, but polynomial if the size of the ballots is randomized.

**Crowdsourcing and social choice** A social choice-theoretic study of collective annotation tasks was done by Kruger et al. [2014] and Qing et al. [2014]. Mechanisms for incentive-compatible elicitation with approval ballots in crowdsourcing applications have been designed by Shah and Zhou [2020]. Meir et al. [2019] define a method to aggregate votes weighted according to their average proximity to the other votes as an estimation of their reliability.

Prelec et al. [2017] introduce the *Bayesian truth serum* approach: eliciting, in addition to the voters' answers, their prediction of the distribution of answers, gives much better results. This approach was generalized by Hosseini et al. [2021] to contexts where the ground truth is a ranking.

Beyond social choice, collective multi-label annotation was first addressed by Nowak and Rüger [2010], who study the agreement between experts and non-experts in some multi-labelling tasks, and by Deng et al. [2014], who solve the multi-label estimation problem with a scalable aggregation method.

## 3 THE MODEL

Let $\mathcal{N} = \{1, \ldots, n\}$ be a set of voters, and $\mathcal{A} = \{a_1, \ldots, a_m\}$ a set of alternatives (possible objects in im-

ages, notes in chords, papers, patients...). Consider a set of $L$ *instances*: an instance $z$ consists of an approval profile $A^z = (A_1^z, \ldots, A_n^z)$ where $A_i^z \subseteq \mathcal{A}$ is an approval ballot for every $i \in \mathcal{N}$. For example, in a crowdsourcing context, a task usually contains multiple questions, and an instance comprises the voters' answers to one of these questions.

For each instance $z \in L$, there exists an *unknown* ground truth $S_z^*$ belonging to $\mathcal{S} = 2^{\mathcal{A}}$, which is the set of objectively correct alternatives in instance $z$. It is prior knowledge by the central authority (but not necessarily by voters), that the number of alternatives in each of them lies in the interval $[l, u]$: $S_z^* \in \mathcal{S}_{l,u} = \{S \in \mathcal{S}, l \leq |S| \leq u\}$, for given bounds $0 \leq l \leq u \leq m$.

Our goal is to unveil the ground truth for each of these instance using the votes and the prior knowledge on the number of winning alternatives. We define a noise model consisting of two parametric distributions, namely, a conditional distribution of the approval ballots given the ground truth, and a prior distribution on the ground truth. Here we depart from classical noise models in epistemic social choice, as we suppose that the parameters of these distributions may be unknown and thus need to be estimated.

For each voter $i \in \mathcal{N}$, we suppose that there exist two unknown parameters $(p_i, q_i)$ in $(0, 1)$ such that the approval ballot $A_i^z$ on an instance $z \in L$ is drawn according to the following distribution: for each $a \in \mathcal{A}$,

$$P(a \in A_i^z | S_z^* = S) = \begin{cases} p_i & \text{if } a \in S \\ q_i & \text{if } a \notin S \end{cases}$$

where $p_i$ (resp. $q_i$) is the (unknown) probability that voter $i$ approves a correct (resp. incorrect) alternative. Then we make the following assumptions:

(1) A voter's approvals of alternatives are mutually independent given the ground truth and parameters $(p_i, q_i)_{i \in \mathcal{N}}$.
(2) Voters' ballots are mutually independent given the ground truth.
(3) Instances are independent given the parameters $(p_i, q_i)_{i \in \mathcal{N}}$ and the ground truths.

To model the prior probability of any set $S$ to be the ground truth $S^*$, we define parameters $t_j = P(a_j \in S^*)$. $t_j$ can be understood as the prior probability of $a_j$ to be in the ground truth set $S^*$ before the cardinality constraints are taken into account. These, together with an independence assumption on the events $\{a_j \in S^*\}$, gives $P(S = S^*) = \prod_{a_j \in S} t_j \prod_{a_j \notin S} 1 - t_j$. Note that the choice of the parameters $t_j$ is not crucial when running the algorithm for estimating the ground truth: we will see in Section 4.3 that it converges whatever their values. The distribution conditional to the prior knowledge on the size of the ground truth $\tilde{P}(S)$ can be seen as a projection on the constraints followed by a normalization:

$$P(S^* = S | l \leq |S^*| \leq u) = \frac{P(S^* = S \cap |S^*| \in [l, u])}{P(|S^*| \in [l, u])}$$

It follows:

$$\tilde{P}(S) = \begin{cases} \frac{1}{\beta(l,u,t)} \prod_{a_j \in S} t_j \prod_{a_j \notin S} (1 - t_j) & \text{if } S \in \mathcal{S}_{l,u} \\ 0 & \text{if } S \notin \mathcal{S}_{l,u} \end{cases}$$

where $\beta(l, u, t) = \sum_{S \in \mathcal{S}_{l,u}} \prod_{a_j \in S} t_j \prod_{a_j \notin S} (1 - t_j)$.

The ground truths associated with different instances are assumed to be mutually independent given the parameters.

Two particular cases are worth discussing. First, when $(l, u) = (0, m)$, the problem is *unconstrained* and we have $\beta(0, m, t) = P(|S^*| \in [0, m]) = 1$, so $\tilde{P}(S) = P(S = S^*)$. In this case the problem degenerates into a series of independent binary label-wise estimations (see Subsection 4.1).

Second, in the single-winner case $(l, u) = (1, 1)$, we have $\tilde{P}(\{a_j\}) = \frac{t_j \prod_{h \neq j} 1 - t_h}{\beta(1,1,t)}$, therefore, for any approval profile $A$:

$$P(S^* = a_j | A) \propto P(A | S = a_j) \tilde{P}(a_j)$$
$$= P(A | S = a_j) \times \frac{t_j \prod_{h \neq j} (1 - t_h)}{\beta}$$
$$= P(A | S = a_j) \times \frac{1}{(1 - t_j)} \frac{t_j \prod_{1 \leq h \leq m} (1 - t_h)}{\beta}$$
$$\propto P(A | S = a_j) \times \frac{t_j}{(1 - t_j)}$$

We recover the same estimation problem if we simply introduce $\alpha_j = P(S^* = \{a_j\})$ with $\sum \alpha_j = 1$ as in Ben-Yashar and Paroush [2001], in which case we have $P(S^* = a_j | A) \propto \alpha_j P(A | S^* = a_j)$.

## 4 ESTIMATING THE GROUND TRUTH

Our aim is the intertwined estimation of the ground truth and the parameters via maximizing the total likelihood of the instances:

$$\mathcal{L}(A, S, p, q, t) = \prod_{z=1}^{L} \tilde{P}(S_z) \prod_{i=1}^{n} P(A_i^z | S_z)$$

where:

$$P(A_i^z | S_z) = p_i^{|A_i^z \cap S_z|} q_i^{|A_i^z \cap \overline{S_z}|} (1 - p_i)^{|\overline{A_i^z} \cap S_z|} (1 - q_i)^{|\overline{A_i^z} \cap \overline{S_z}|}$$

To this aim, we will introduce an iterative algorithm whose main two steps will be presented in sequence, in the next subsections, before the main algorithm is formally defined and its convergence shown. These two steps are:

- Estimating the ground truths given the parameters.
- Estimating the parameters given the ground truths.

Simply put, the algorithm consists in iterating these two steps until it converges to a fixed point.

## 4.1 ESTIMATING THE GROUND TRUTH GIVEN THE VOTES AND THE PARAMETERS

Since instances are independent given the parameters, we focus here on one instance with ground truth $S^*$ and profile $A = (A_1, \ldots, A_n)$. Before diving into maximum likelihood estimation (MLE), we introduce some notions and prove some lemmas. In this subsection, we suppose that the parameters $(p_i, q_i)_{i \in \mathcal{N}}$ and $(t_j)_{j \in \mathcal{A}}$ are known (later on, these parameters will be replaced by their estimations at each iteration of the algorithm). Thus, all in all, input and output are as follows:

- Input: approval profile $A$; parameters $(p_i, q_i)_{i \in \mathcal{N}}$ and $(t_j)_{j \in \mathcal{A}}$.
- Output: MLE of the ground truth $S^*$.

**Definition 1** (weighted approval score). *Given an approval profile* $(A_1, \ldots, A_n)$, *noise parameters* $(p_i, q_i)_{1 \leq i \leq n}$ *and prior parameters* $(t_j)_{1 \leq j \leq m}$, *define:*

$$app_w(a_j) = ln\left(\frac{t_j}{1 - t_j}\right) + \sum_{i: a_j \in A_i} ln\left(\frac{p_i(1 - q_i)}{q_i(1 - p_i)}\right)$$

The scores $app_w(a_j)$ can be interpreted as weighted approval scores for a $(n + m)$-voter profile where:

- for each voter $1 \leq i \leq n$: $i$ has a weight $w_i = ln\left(\frac{p_i(1-q_i)}{q_i(1-p_i)}\right)$ and casts approval ballot $A_i$.
- for each $1 \leq j \leq m$: there is a virtual voter with weight $w_j = ln\left(\frac{t_j}{1-t_j}\right)$ who casts approval ballot $A_j = \{a_j\}$.

While the weight of each voter $i \in \mathcal{N}$ depends on her reliability, each prior information on an alternative plays the role of a virtual voter who only selects the concerned alternative, with a weight that increases as the prior parameter increases.

From now on, we suppose without loss of generality that the alternatives are ranked according to their score:

$$app_w(a_1) \geq app_w(a_2) \geq \cdots \geq app_w(a_m)$$

**Definition 2** (threshold and partition). *Define the threshold:*

$$\tau_n = \sum_{i=1}^{n} ln\left(\frac{1 - q_i}{1 - p_i}\right)$$

*and the partition of the set of alternatives in three sets:*

$$\begin{cases} S_{max}^{\tau_n} &= \{a \in A, app_w(a) > \tau_n\} \\ S_{tie}^{\tau_n} &= \{a \in A, app_w(a) = \tau_n\} \\ S_{min}^{\tau_n} &= \mathcal{A} \backslash (S_{max}^{\tau_n} \cup S_{tie}^{\tau_n}) \end{cases}$$

*and let* $k_{max}^{\tau_n} = |S_{max}^{\tau_n}|, k_{tie}^{\tau_n} = |S_{tie}^{\tau_n}|, k_{min}^{\tau_n} = |S_{min}^{\tau_n}|$.

The next result characterizes the sets in $\mathcal{S}$ that are MLEs of the ground truth given the parameters.

**Theorem 1.** $\tilde{S} \in \arg\max_{S \in \mathcal{S}} \mathcal{L}(A, S, p, q, t)$ *if and only if there exists* $k \in [l, u]$ *such that* $\tilde{S}$ *is the set of* $k$ *alternatives with the highest* $k$ *values of* $app_w$ *and:*

$$\begin{cases} |\tilde{S} \cap S_{max}^{\tau_n}| &= \min(u, k_{max}^{\tau_n}) \\ |\tilde{S} \cap S_{min}^{\tau_n}| &= \max(0, l - k_{tie}^{\tau_n} - k_{max}^{\tau_n}) \end{cases} \quad (1)$$

So the estimator $\tilde{S}$ is made of some top-$k$ alternatives, where the possible values of $k$ are determined by Eq. (1). The first equation imposes that $\tilde{S}$ includes as many elements as possible from $S_{max}^{\tau_n}$ (without exceeding the upper-bound $u$), whereas the second one imposes that $\tilde{S}$ includes as few elements as possible from $S_{min}^{\tau_n}$ (without getting below the lower-bound $l$). An example is included in the appendix.

*Proof.* Since $\tilde{P}(S) > 0 \iff S \in \mathcal{S}_{l,u}$, we have that $\arg\max_{S \in \mathcal{S}} \mathcal{L}(S) = \arg\max_{S \in \mathcal{S}_{l,u}} \mathcal{L}(S)$. Moreover, we have that for any $S \in \mathcal{S}_{l,u}$:

$$\mathcal{L}(S) = \tilde{P}(S) \prod_{i=1}^{n} p_i^{|A_i \cap S|} q_i^{|A_i \cap \overline{S}|} (1 - p_i)^{|\overline{A_i} \cap S|} (1 - q_i)^{|\overline{A_i} \cap \overline{S}|}$$

$$= \tilde{P}(S) \prod_{i=1}^{n} p_i^{|A_i \cap S|} q_i^{|A_i| - |A_i \cap S|} (1 - p_i)^{|S| - |A_i \cap S|}$$

$$(1 - q_i)^{|\overline{A_i}| - |S| + |A_i \cap S|}$$

$$\propto \tilde{P}(S) \prod_{i=1}^{n} \left[\frac{1 - p_i}{1 - q_i}\right]^{|S|} \left[\frac{p_i(1 - q_i)}{q_i(1 - p_i)}\right]^{|A_i \cap S|}$$

$$\propto \frac{1}{\beta} \prod_{a_j \in S} t_j \prod_{a_j \notin S} (1 - t_j) \prod_{i=1}^{n} \left[\frac{1 - p_i}{1 - q_i}\right]^{|S|} \left[\frac{p_i(1 - q_i)}{q_i(1 - p_i)}\right]^{|A_i \cap S|}$$

$$\propto \prod_{a_j \in S} \frac{t_j}{1 - t_j} \prod_{i=1}^{n} \left[\frac{1 - p_i}{1 - q_i}\right]^{|S|} \left[\frac{p_i(1 - q_i)}{q_i(1 - p_i)}\right]^{|A_i \cap S|}$$

Thus the log-likelihood $\ell(S)$ reads:

$$\sum_{a_j \in S} \ln \frac{t_j}{1 - t_j} + \sum_{i=1}^{n} |S| \ln \frac{1 - p_i}{1 - q_i} + |A_i \cap S| \ln \frac{p_i(1 - q_i)}{q_i(1 - p_i)}$$

$$= \sum_{a_j \in S} \left[ \underbrace{\underbrace{\ln \frac{t_j}{1 - t_j} + \sum_{i: a_j \in A_i} \ln \frac{p_i(1 - q_i)}{q_i(1 - p_i)}}_{app_w(a_j)} }_{l(a_j)} - \underbrace{\sum_{i=1}^{n} \ln \frac{1 - q_i}{1 - p_i}}_{\tau_n} \right]$$

This means that $a \in S_{max}^{\tau_n}$ if and only if $\ell(a) > 0$, $a \in S_{min}^{\tau_n}$ if and only if $\ell(a) < 0$ and $a \in S_{tie}^{\tau_n}$ if and only if $\ell(a) = 0$. Now, let $S_M$ be a maximizer of the likelihood. Since $\ell(a_j) \geq \ell(a_h) \iff app_w(a_j) \geq app_w(a_h)$ we have that $S_M$, which maximizes $\sum_{a_j \in S} \ell(a_j)$, is made of top-$k$ alternatives for some $k \in [l \mathinner{\ldotp\ldotp} u]$.

Furthermore, $|S_M \cap S_{min}^{\tau_n}| = \max(0, l - k_{tie}^{\tau_n} - k_{max}^{\tau_n})$. Start by noticing that $|S_M \cap S_{min}^{\tau_n}| \geq \max(0, l - k_{tie}^{\tau_n} - k_{max}^{\tau_n})$,

since $|S_M \cap S_{min}^{\tau_n}| \geq l - |S_M \cap S_{max}^{\tau_n}| - |S_M \cap S_{tie}^{\tau_n}| \geq l - k_{max}^{\tau_n} - k_{tie}^{\tau_n}$. Suppose that $|S_M \cap S_{min}^{\tau_n}| > \max(0, l - k_{max}^{\tau_n} - k_{tie}^{\tau_n})$. Then we have that $|S_M| > l$ because otherwise, if $|S_M| = l$, then $|S_M \cap S_{max}^{\tau_n}| + |S_M \cap S_{tie}^{\tau_n}| = l - |S_M \cap S_{min}^{\tau_n}| < k_{max}^{\tau_n} + k_{tie}^{\tau_n}$, which would mean that there are elements in $S_{tie}^{\tau_n}$ and $S_{max}^{\tau_n}$ which are not in $S_M$, which is a contradiction since $|S_M \cap S_{min}^{\tau_n}| > 0$ and $S_M$ is a top-$k$ set. Now consider $a \in S_M \cap S_{min}^{\tau_n}$, we have that $|S_M \backslash \{a\}| \geq l$ and $l(S_M) = l(S_M \backslash \{a\}) + l(a) < l(S_M \backslash \{a\})$ which is a contradiction.

With the same idea we can prove that $|S_M \cap S_{max}^{\tau_n}| = \min(u, k_{max}^{\tau_n})$.

Conversely, consider an admissible set $S$ of top-$k$ alternatives that verifies the constraints (1). Let $S_M$ be a MLE which, by the first part of the proof, is a top-$k'$ set that also satisfies the same constraints (1). Thus we have that $|S_M \cap S_{max}^{\tau_n}| = |S \cap S_{max}^{\tau_n}| = \min(u, k_{max}^{\tau_n})$, and since $S$ and $S_M$ are top-$k$ and top-$k'$ sets, we have that $S \cap S_{max}^{\tau_n} = S_M \cap S_{max}^{\tau_n}$. Similarly we have that $S \cap S_{min}^{\tau_n} = S_M \cap S_{min}^{\tau_n}$. This suffices to prove that $\ell(S) = \ell(S_M)$ is maximal. $\square$

Notice that when $(l, u) = (0, m)$, the problem degenerates into a collection of label-wise problems, one for each alternative: $a_j$ is selected if $a_j \in S_{max}^{\tau_n}$, rejected if $a_j \in S_{min}^{\tau_n}$, and those that are on the fence can be arbitrarily selected or not.

**Example 1.** *Consider* 5 *alternatives* $\mathcal{A} = \{a, b, c, d, e\}$ *and* 10 *voters* $\mathcal{N}$ *all sharing the same parameters* $(p, q) = (0.7, 0.4)$. *We thus have that all voters share the same weight* $w = ln\left(\frac{p(1-q)}{q(1-p)}\right) = 1.25$ *and* $\tau_n = \sum_{i=1}^n ln\left(\frac{1-q}{1-p}\right) = 6.93$. *We consider the constraints* $(l, u) = (1, 4)$

*First, suppose that* $t_d = 0.6$ *and that* $t_j = 0.5$ *for all the remaining candidates. Consider also the approval counts (and weighted approval scores) in the table below.*

| Candidate | a | b | c | d | e |
|-----------|------|----|------|------|------|
| Approval count | 9 | 8 | 7 | 5 | 5 |
| $app_w$ | 11.25 | 10 | 8.75 | 6.65 | 6.25 |

*We can easily check, by Theorem* 1 *that* $\tilde{S} = \arg\max_{S \in \mathcal{S}} P(S = S^*|A) = \{a, b, c\}$. *We have that* $S_{max}^{\tau_n} = \{a, b, c\}, S_{tie}^{\tau_n} = \emptyset$ *and* $S_{min}^{\tau_n} = \{d, e\}$. *We know that there exists some* $k \in [1, 4]$ *such that* $\tilde{S}$ *would consist of the top* $k$ *alternatives. We also have that:*

$$\begin{cases} |\tilde{S} \cap S_{max}^{\tau_n}| &= \min(u, k_{max}^{\tau_n}) = 3 \implies \{a, b, c\} \subseteq \tilde{S} \\ |\tilde{S} \cap S_{min}^{\tau_n}| &= \max(0, l - k_{tie}^{\tau_n} - k_{max}^{\tau_n}) = 0 \implies d, e \notin \tilde{S} \end{cases}$$

*So the only possibility is* $\tilde{S} = \{a, b, c\}$.

## 4.2 ESTIMATING THE PARAMETERS GIVEN THE GROUND TRUTH

### 4.2.1 Estimating the prior parameters over alternatives

Once the ground truths are estimated at one iteration of the algorithm, the next step consists in estimating the prior parameters $(t_j)_{j \in \mathcal{A}}$, with the ground truths being given (in Subsection 4.3 the ground truth will be replaced by its estimation at each iteration). The next proposition explicits the closed-form expression of the MLE of the prior parameter of each alternative given the ground truth of each instance $S_z^*$ once the prior parameters of all other alternatives are fixed.

- Input: Approval profile $(A_1, \ldots, A_n)$, ground truths $S_z^*$, and all but one prior parameters $(t_h)_{h \neq j}$.
- Output: MLE of $t_j$.

**Proposition 2.** *For every* $a_j \in \mathcal{A}$:

$$\arg\max_{t \in (0,1)} \mathcal{L}(A, S, p, q, t, t_{-j}) = \frac{occ(j)\overline{\alpha}_j}{(L - occ(j))\underline{\alpha}_j + occ(j)\overline{\alpha}_j}$$

$$where: \begin{cases} \overline{\alpha}_j &= \sum_{\substack{S \in \mathcal{S}_{l,u} \\ a_j \in S}} \prod_{\substack{a_h \in S \\ h \neq j}} t_h \prod_{a_h \notin S} (1 - t_h) \\ \underline{\alpha}_j &= \sum_{\substack{S \in \mathcal{S}_{l,u} \\ a_j \notin S}} \prod_{a_h \in S} t_h \prod_{\substack{a_h \notin S \\ h \neq j}} (1 - t_h) \\ occ(j) &= |z \in \{1, \ldots, L\}, a_j \in S_z| \end{cases}$$

Notice that $\overline{\alpha}_j = P(l \leq |S^*| \leq u|a_j \in S^*)$ and $\underline{\alpha}_j = P(l \leq |S^*| \leq u|a_j \notin S^*)$ so $\beta = \overline{\alpha}_j t_j + \underline{\alpha}_j(1 - t_j)$. $occ(j)$ is the number of instances whose ground truth contains $a_j$.

*Proof.* Fix all sets $S_z \in \mathcal{S}_{l,u}$ and all the noise parameters $(p_i, q_i)_i$ and all the prior parameters $t_h$ but for one $t_j$ for some $j \leq m$, and let $t \in (0, 1)$:

$$\mathcal{L}(S, t, t_{-j}) \propto \prod_{z=1}^L \frac{1}{\beta(l, u, t)} \prod_{a_h \in S_z} t_h \prod_{a_h \notin S_z} (1 - t_h)$$

$$\propto \prod_{z=1}^L \frac{1}{\beta(l, u, t, t_{-j})} \prod_{a_h \in S_z} t_h \prod_{a_h \notin S_z} (1 - t_h)$$

$$\propto \left(\frac{1}{\beta(l, u, t, t_{-j})}\right)^L \underbrace{\prod_{z:a_j \in S_z} t}_{t^{occ(j)}} \underbrace{\prod_{z:a_j \notin S_z} (1 - t)}_{(1-t)^{L-occ(j)}}$$

Taking the log we can write the function as:

$$\ell(t) = -L \log \beta + occ(j) \log t + (L - occ(j)) \log(1 - t)$$

Its derivative reads:

$$\frac{\partial \ell}{\partial t} = -L \frac{\underline{\alpha}_j - \overline{\alpha}_j}{\underline{\alpha}_j t + \overline{\alpha}_j(1 - t)} + occ(j)\frac{1}{t} + (occ(j) - L)\frac{1}{1 - t}$$

Canceling it, we obtain:

$$t = \frac{occ(j)\overline{\alpha}_j}{(L - occ(j))\underline{\alpha}_j + occ(j)\overline{\alpha}_j}$$

The derivative vanishes in a single point in $(0, 1)$ and $\lim_{t\to 0} \ell(t) = \lim_{t\to 1} \ell(t) = -\infty$ thus $\ell$ reaches a unique maximum. $\qquad\square$

We will see later that the algorithm applies Proposition 2 sequentially to estimate the alternatives' parameters one by one (see Example 2).

### 4.2.2 Estimating the voter parameters

Once the ground truths are known (or estimated), we can estimate the voters' parameters $(p, q)$.

- Input: Instances $(A^1, \ldots, A^L)$, ground truths $(S_1^*, \ldots, S_L^*)$.
- Output: MLE of voter reliabilities $(p, q)$.

The next result simply states that the maximum likelihood estimator of $p_i$ of some voter is the fraction of alternatives that the voter approves and that actually belong to the ground truth; the estimation of $q_i$ is similar. See Example 2.

**Proposition 3.** *Fix sets $S_z \in \mathcal{S}_{l,u}$ and prior parameters $t_j$. Then:*

$$\underset{(p,q)\in(0,1)^{2n}}{\arg\max} \mathcal{L}(A, S, p, q, t) = (\hat{p}, \hat{q})$$

*where:* $\hat{p}_i = \frac{\sum_{z\in L} |A_i^z \cap S_z|}{\sum_{z\in L} |S_z|}, \hat{q}_i = \frac{\sum_{z\in L} |A_i^z \cap \overline{S_z}|}{\sum_{z\in L} |\overline{S_z}|}$

The (simple) proof is omitted.

### 4.3 ALTERNATING MAXIMUM LIKELIHOOD ESTIMATION

Now the estimation of the ground truths and that of the parameters are intertwined to maximize the overall likelihood $\mathcal{L}(A, S, p, q, t)$ by the *Alternating Maximum Likelihood Estimation algorithm*. AMLE is an iterative procedure similar to the *Expectation-Maximization* procedure introduced in Baharad et al. [2011] but with a coordinate-steepest-ascent-like iteration, whose aim is to intertwinedly estimate the voter reliabilities, the alternatives' prior parameters and the instances' ground truths. The idea behind this estimation consists in alternating a MLE of the ground truths given the current estimate of the parameters, and an updating of these parameters via a MLE based on the current estimate of the ground truths.[1] Each of these steps have been discussed

---

[1] In case of ties between subsets when estimating the ground truth, a tie-breaking priority over subsets is used. No ties occurred in our experiments.

---

**Algorithm 1** *AMLE* procedure

**Input:** Approval ballots $(A_i^z)_{1\le z\le L, i\in\mathcal{N}}$
  Initial parameters $\hat{\theta}^{(0)}$, Bounds $(l, u)$, error $\varepsilon$
**Output:** Estimations $(\hat{S}_z), (\hat{p}_i, \hat{q}_i), (\hat{t}_j)$
**repeat**
  **for** $z = 1 \ldots L$ **do**
    Compute $\hat{S}_z^{(v+1)} = \{a_1, \ldots, a_k\}$ with $k \in [l, u]$
    and:

$$\begin{aligned}
|\hat{S}_z^{(v+1)} \cap S_{max,z}^{(v)}| &= \min(u, k_{max,z}^{(v)}) \\
|\hat{S}_z^{(v+1)} \cap S_{min,z}^{(v)}| &= \max(0, l - k_{tie,z}^{(v)} - k_{max,z}^{(v)})
\end{aligned}$$

  **end for**
  **for** $i = 1 \ldots \mathcal{N}$ **do**
    Update the parameters $(p_i, q_i)$ given $\hat{S}^{(v+1)}$:

$$\hat{p}_i^{(v+1)} = \frac{\sum\limits_{z\in L} |A_i^z \cap \hat{S}_z^{(v+1)}|}{\sum\limits_{z\in L} |\hat{S}_z^{(v+1)}|}, \hat{q}_i^{(v+1)} = \frac{\sum\limits_{z\in L} |A_i^z \cap \overline{\hat{S}_z^{(v+1)}}|}{\sum\limits_{z\in L} |\hat{S}_z^{(v+1)}|}$$

  **end for**
  **for** $j = 1 \ldots m$ **do**
    Update $\hat{t}_j^{(v+1)}$ by:

$$\hat{t}_j^{(v+1)} = \frac{occ^{(v+1)}(j)\overline{\alpha}_j^{(v+1)}}{occ^{(v+1)}(j)\overline{\alpha}_j^{(v+1)} + (L - occ^{(v+1)}(j))\underline{\alpha}_j^{(v+1)}}$$

    where :

$$\begin{cases}
occ^{(v+1)}(j) &= \sum_{z=1}^{L} \mathbb{1}\{a_j \in \hat{S}_z^{(v+1)}\} \\
\overline{\alpha}_j^{(v+1)} &= \beta((l-1)^+, u-1, \hat{t}_{<j}^{(v+1)}, \hat{t}_{>j}^{(v)}) \\
\underline{\alpha}_j^{(v+1)} &= \beta(l, u, \hat{t}_{<j}^{(v+1)}, \hat{t}_{>j}^{(v)})
\end{cases}$$

  **end for**
**until** $||\hat{\theta}^{(v+1)} - \hat{\theta}^{(v)}|| \le \varepsilon$

---

in the previous subsections and are now incorporated into Algo. 1.

The algorithm continues to run until a convergence criterion is met in the form of a bound on the norm of the change in the parameters' estimations. In practice we chose $\ell_\infty$, but any other norm could be used in Algorithm 1 as in finite dimensions, all norms are equivalent (if a sequence converges according to one norm then it does so for any norm).

We define the vector of parameters $\hat{\theta}^{(v)} = (\hat{p}^{(v)}, \hat{q}^{(v)}, \hat{t}^{(v)})$ containing the voters' estimated noise parameters as well as the prior information estimated parameters at iteration $v$. In particular $\hat{\theta}^{(0)}$ is the input initial values. The choice of the exact initial values depends on the application at hand.

Note that at convergence, only local optimality is guaranteed.

**Theorem 4.** *For any initial values $\hat{\theta}^{(0)}$, AMLE converges to a fixed point after a finite number of iterations.*

We only provide a sketch of proof and defer the full proof to the Appendix.

*Proof.* First we have by Theorem 1 that:

$$\mathcal{L}(A, \hat{S}^{(v+1)}, \hat{\theta}^{(v)}) \geq \mathcal{L}(A, \hat{S}^{(v)}, \hat{\theta}^{(v)})$$

By Proposition 2 and Proposition 3, we deduce that:

$$\mathcal{L}(A, \hat{S}^{(v+1)}, \hat{\theta}^{(v+1)}) \geq \mathcal{L}(A, \hat{S}^{(v+1)}, \hat{\theta}^{(v)})$$

Hence, the likelihood increases at every step. Since there is a finite number of possible values for the ground truth (namely $2^{mL}$), the convergence of the algorithm is guaranteed. □

Because $\mathcal{L}(A, \hat{S}^{(v+1)}, \hat{\theta}^{(v+1)}) \geq \mathcal{L}(A, \hat{S}^{(v+1)}, \hat{\theta}^{(v)}) \geq \mathcal{L}(A, \hat{S}^{(v)}, \hat{\theta}^{(v)})$, the likelihood increases at each step of the algorithm. This guarantees that whenever the execution stops, the likelihood is closer to the maximum than it initially was. Therefore the algorithm can not only be run until convergence, but it can also be run as an anytime algorithm.

**Example 2.** *Take $n = 3$, $m = 5$, $l = 1$, $u = 2$, $L = 4$, and the following profile and initial parameters:*

|         | $A^1$          | $A^2$              | $A^3$          | $A^4$   |
|---------|----------------|--------------------|----------------|---------|
| Voter 1 | $\{a_1, a_4\}$ | $\{a_1\}$          | $\{a_3\}$      | $\{a_1\}$ |
| Voter 2 | $\{a_2\}$      | $\{a_5\}$          | $\{a_4\}$      | $\{a_1\}$ |
| Voter 3 | $\{a_2, a_3, a_4\}$ | $\{a_2, a_3, a_5\}$ | $\{a_2, a_3\}$ | $\{a_3\}$ |

$$\begin{cases} \hat{p}_1^{(0)} = 0.5 & \hat{p}_2^{(0)} = 0.5 & \hat{p}_3^{(0)} = 0.5 \\ \hat{q}_1^{(0)} = 0.44 & \hat{q}_2^{(0)} = 0.41 & \hat{q}_3^{(0)} = 0.32 \\ \hat{t}_1^{(0)} = \cdots = \hat{t}_5^{(0)} & = 0.5 \end{cases}$$

*Estimating the ground truth:* *The first step is the application of Theorem 1 to estimate the ground truth of the instances given the initial parameters, yielding $\hat{S}_1^{(1)} = \{a_2, a_4\}, \hat{S}_2^{(1)} = \{a_2, a_5\}, \hat{S}_3^{(1)} = \{a_2, a_3\}, \hat{S}_4^{(1)} = \{a_1, a_3\}$*

*Estimating the voter reliabilities:* *In the next step we use these estimates of the ground truths to compute the MLEs of the voter reliabilities. For instance, voter 1 has 2 false positive labels from a total of 12 negative labels so $\hat{q}_1^{(1)} = \frac{2}{12} = 0.17$ and she has 3 true positive labels out of 8 positive ones so $\hat{p}_1^{(1)} = \frac{3}{8} = 0.38$. In the end, we get:*

$$\begin{cases} \hat{p}_1^{(1)} = 0.38 & \hat{p}_2^{(1)} = 0.38 & \hat{p}_3^{(1)} = 0.88 \\ \hat{q}_1^{(1)} = 0.17 & \hat{q}_2^{(1)} = 0.08 & \hat{q}_3^{(1)} = 0.17 \end{cases}$$

*Estimating the prior parameters:* *The final step of this iteration consists in updating the estimations of the prior parameters by applying Proposition 2 sequentially. First we estimate $\hat{t}_1^{(1)}$ given $\hat{S}^{(1)}$ and $\hat{t}_2^{(0)}, \ldots, \hat{t}_5^{(0)}$ by maximum likelihood estimation. We first compute $\overline{\alpha}_1 = \beta(0, 1, t_2, \ldots, t_5) = 0.3125$, $\underline{\alpha}_1 = \beta(1, 2, t_2, \ldots, t_5) = 1$ and $occ(a_1) = 1$. Then the MLE of $t_1$ is:*

$$\hat{t}_1 = \frac{occ(a_1)\overline{\alpha}_1}{(L - occ(a_1))\underline{\alpha}_1 + occ(a_1)\overline{\alpha}_1} = 0.09$$

*The next steps are to estimate $\hat{t}_2^{(1)}$ given $\hat{t}_1^{(1)}, \hat{t}_3^{(0)}, \hat{t}_4^{(0)}, \hat{t}_5^{(0)}$ and so on. Finally, we get:*

$$\hat{t}_1^{(1)} = 0.09, \hat{t}_2^{(1)} = 0.56, \hat{t}_3^{(1)} = 0.28, \hat{t}_4^{(1)} = 0.14, \hat{t}_5^{(1)} = 0.20$$

*Fix $\varepsilon = 10^{-5}$. We repeat all steps until convergence (according to $\ell_\infty$), after 5 full iterations. In the fixed point, the estimations of the ground truths are:*

$$\hat{S}_1 = \{a_2, a_3\}, \hat{S}_2 = \{a_2, a_3\}, \hat{S}_3 = \{a_2, a_3\}, \hat{S}_4 = \{a_3\}$$

# 5 EXPERIMENTS

## 5.1 EXPERIMENT DESIGN AND DATA COLLECTION

We designed an image annotation task as a football quiz.[2] We selected 15 pictures taken during different matches between two of the following teams: Real Madrid, Inter Milan, Bayern Munich, Barcelona, Paris Saint-Germain. In each picture, it may be the case that players from both teams appear, or players from only one team, therefore $l = 1$ and $u = 2$. Each participant is shown the instances one by one, and is each time asked to select all the teams she can spot (see Figure 1). We designed a simple incentive for participants, consisting in ranking them according to the following principle:

- The participants get one point whenever their answer contains all correct alternatives for a picture. They are then ranked according to their cumulated points.
- To break ties, the participant who selected a smaller number of alternatives overall is ranked first.

We gathered the answers of 76 participants: only two of them spammed by simply selecting all the alternatives. Figure 2 shows that voters responded well to the incentives by mostly selecting one or two alternatives.

## 5.2 ANNA KARENINA'S INITIALIZATION

Inspired by the *Anna Karenina Principle* in Meir et al. [2019], we assign more weight to voters who are *closer*

---

[2]The dataset and code are accessible at https://github.com/taharallouche/Football-Quiz-Crowdsourcing

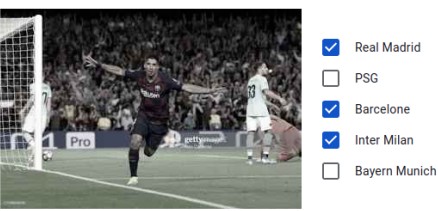

Figure 1: Example of Annotation Task

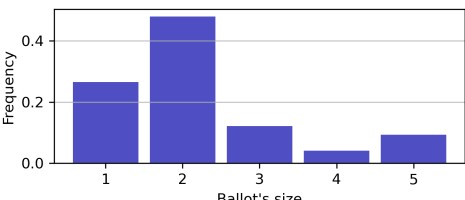

Figure 2: Histogram of answers' size

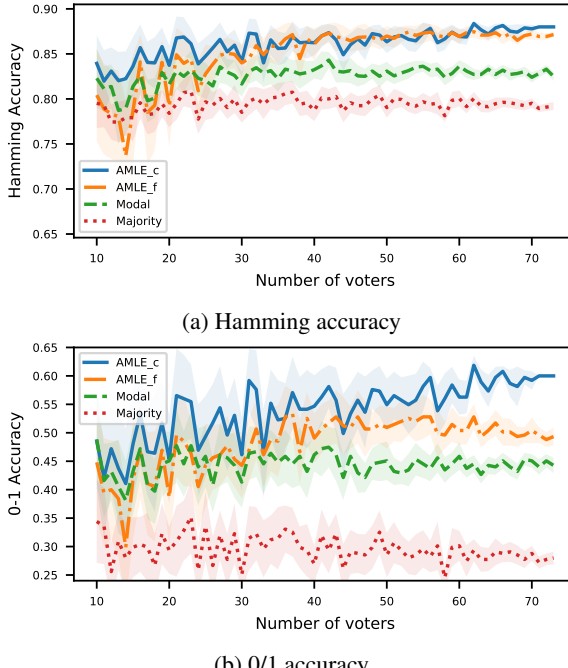

(a) Hamming accuracy

(b) 0/1 accuracy

Figure 3: Accuracies of different aggregation methods

to the others on average, initializing the precision parameters $(p_i, q_i)$ accordingly. This suits our context, where voter competence is highly polarized: some voters are experts and cast similar answers close to the ground truth, the others are less reliable and their answers are dispersed among all combinations.

We use the following heuristics (see Algorithm 2) for the initialization:

---

**Algorithm 2** Initializing $(p_i, q_i)_i$

**Input:**     Approval ballots $(A_i^z)_{z,i}$
**Output:**   Initialization $(\hat{p}_i^{(0)}, \hat{q}_i^{(0)})$

-Compute $w_{max} = \frac{n}{1+n}, w_{min} = \frac{1}{1+n}$
-Compute $d_i = \sum_{j \neq i} d_{Jacc}(A_i, A_j)$ (Jaccard distance)
-Compute $d_{max} = \max d_i, d_{min} = \min d_i$
-Compute $w_i = (w_{max} - w_{min}) \left( \frac{1/d_i - 1/d_{max}}{1/d_{min} - 1/d_{max}} \right) + w_{min}$
-Fix $\hat{p}_i^{(0)} = \frac{1}{2}$ and $\hat{q}_i^{(0)} = \frac{1 - \frac{e^{w_i} - 1}{e^{w_i} + 1}}{2}$

---

Algorithm 2 guarantees that the parameters $(\hat{p}_i^{(0)}, \hat{q}_i^{(0)})$ of a voter are such that her initial weight is equal to $w_i$, and that $\frac{w_{max}}{w_{min}} = n$: therefore, initially, the voter closest in average to the other voters counts $n$ times more than the voter with the largest average distance.

In the Appendix we give an example illustrating this initialization, and an empirical comparison with other classical initializations.

## 5.3   RESULTS

To assess the importance of prior information on the size of the ground truth, we tested the AMLE algorithm with free bounds $(l, u) = (0, m)$ (will be referred to as AMLE$_f$) and the AMLE$_c$ algorithm with $(l, u) = (1, 2)$. We also apply the modal rule Caragiannis et al. [2020] which outputs the subset of alternatives that most frequently appears as an approval ballot $\arg \max_{S \in \mathcal{S}} |i \in \mathcal{N}, S = A_i|$, and a variant of label-wise majority rule which outputs the subset of alternatives $S$ such that $a \in S \iff |i \in \mathcal{N}, a \in A_i| > \frac{n}{2}$. If this subset is empty it is replaced by the alternative with highest approval count, and if it has more than two alternatives then we only keep the top-2 alternatives.

We took 20 batches of $n = 10$ to $n = 74$ randomly drawn voters and applied the four methods to all of them (see Figure 3a,3b). As classically done in the literature Nguyen et al. [2020], we use the Hamming accuracy $\frac{1}{mL} \sum_{z=1}^{L} |S_z^* \cap \hat{S}^z| + |\overline{S_z^*} \cap \overline{\hat{S}^z}|$ and the 0/1 accuracy $\frac{1}{L} \sum_{z=1}^{L} \mathbb{1}\{S_z^* = \hat{S}^z\}$ as metrics and report their 0.95 confidence intervals.

We notice that the majority and the modal rule are outperformed by AMLE, which can be explained by the fact that they do not take into account the voters' reliabilities. Comparing the performances of AMLE$_c$ and AMLE$_f$ emphasizes the importance of the prior knowledge on the committee size to improve the quality of the estimation.

We also compared the execution time of AMLE$_c$ and AMLE$_f$ (see Figure 4) when run on Intel Core i7-10610U CPU @1.80Ghz 4 cores, 8 threads and 32Gb RAM. Unsur-

prisingly, AMLE$_c$ needs more running time, especially for more than 40 voters.

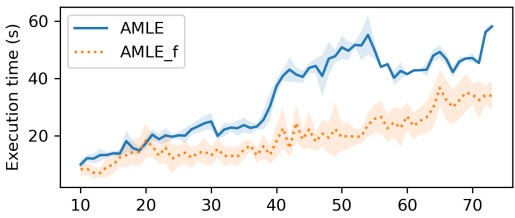

Figure 4: Execution time

# 6 CONCLUSION

We study multi-winner approval voting from an epistemic point of view. The specificity of our work is threefold: (a) the ground truth consists of a set of alternatives; (b) the input consists of approval votes; (c) the competence of the various voters is not known *a priori* but learnt from the input. We proposed a noise model that incorporates the prior belief about the size of the ground truth. Then we derived an iterative algorithm to intertwinedly estimate the ground truth labels, the voter noise parameters and the prior belief parameters and we prove its convergence. Our algorithm is based on a simplification of Expectation-Maximization (EM), and its simple steps are more easily explainable to voters than EM and other similar statistical learning approaches.

Although we mainly considered a general multi-instance task that fits the collective annotation framework, where each voter answers several questions on the same set of alternatives, we can nonetheless apply the same algorithm to single-instance problems (such as the allocation of scarce medical resources) where only one question is answered. In this case, the prior parameters cannot be updated and it suffices to fix them once and for all and alternate between the estimation of the ground truth and the voter parameters.

In some contexts (*e.g.*, patients in a hospital), alternatives and votes are not observed at once but streamed. To cope with this online setup we consider extending our AMLE algorithm in the spirit of Cappé and Moulines [2009].

## Acknowledgements

This work was funded in part by the French government under management of Agence Nationale de la Recherche as part of the "Investissements d'avenir" program, reference ANR-19-P3IA-0001 (PRAIRIE 3IA Institute).

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
