# OpenReview forum: "Multi-winner Approval Voting Goes Epistemic"
_auai.org/UAI/2022/Conference — UAI 2022 Poster_

### Official Review · Reviewer_1f9L · 2022-04-07

**Q2(1) Originality/Novelty:** 3
**Q2(2) Significance/Impact:** 3
**Q2(3) Correctness/Technical Quality:** 2
**Q2(6) Clarity Of Writing:** 3
**Q6 Overall Score:** 6
**Q8 Confidence In Your Score:** 2

**Q1 Summary And Contributions:**

The paper proposes a mechanism for combining approval ballots in the epistemic voting. In the voting, each voter choose a subset of alternatives that he/she believes to  be the ground truth. The aggregation mechanism is an alternating maximal likelihood estimation (AMLE). A small-scaled experiments show that the proposed mechanism outperforms simple majority or modal rules.

**Q2 Assessment Of The Paper:**

More detailed information regarding each of these aspects is given below:

**Q2(4) Quality Of Experiments (Optional):**

3: Good: The experimental evaluation is adequate, and the results convincingly support the main claims.

**Q2(5) Reproducibility:**

3: Good: Key resources (e.g., proofs, code, data) are available and key details (e.g., proofs, experimental setup) are sufficiently well-described for competent researchers to confidently reproduce the main results.

**Q3 Main Strengths:**

The formulation of the aggregation problem in multi-winner epistemic voting as an alternating maximal likelihood estimation is a novel idea. The experiments show that it can uncover the ground truth more accurately than simple majority or modal rules. The paper is mostly well-written and easy to follow except some minor details.

**Q4 Main Weakness:**

The paper appears technically sound, but it lack enough details in some places where more explicit derivations are needed.

**Q5 Detailed Comments To The Authors:**

While the paper is mostly well-presented, there remain some technical questions to be clarified as follows.

1. In Section 3, it is claimed that the choice of the parameters t_j is not crucial when running the algorithm for estimating the ground truth and the reason is that the algorithm converges whatever their values. However, as mentioned in the paper, the AMLE algorithm will only converge to a local optima. Then, even though the choice of prior probability does not matter in terms of convergence, is it still possible that the choice has impact on the final optimal values? In addition, because of the normalization factor for calculating \tilde{P}, all t_j's must be in the open interval (0,1). This seemingly excludes the possibility of modeling partial prior knowledge, i.e., for some alternative j that is known to be in or not in the ground truth in advance. It seems that the best way to model such situations is to set t_j=1 or 0.

2. In the last paragraph of Section 3, I cannot see why P(S*={a_j}|A, |S*|=1} is proportional to rhs of both equations. If it is true, then t_j/(1-t_j) is proportional to \alpha_j, which by definition is t_j\cdot\Pi_{i\not=j}(1-t_i). As all t_j's are independently given, I cannot see any relationship between  \Pi_{i\not=j}(1-t_i) and 1/(1-t_j) . I think more explicit derivation is need here to justify the correctness of these two equations. The same occurs in the proof of Theorem 1 (the derivation of the fifth equation from the forth one). Besides, the first statement in the proof depends on the hidden assumption that t_j is not equal to 1 or 0. Otherwise the implication from S\in S_{l,u} to \tilde{S}>0 does not hold.

3. The symbol L is overloading. It sometimes mean the set of instances and sometimes mean the likelihood estimation (in the proof of Theorem 1). Furthermore, the likelihood function is written as L(S), l(s) (both in the proof) and sometimes as {\cal L} with all arguments. I suggest the authors to avoid such notational confusions.

4. It is claimed that the proof of Proposition 2 is in the Appendix, but I cannot find it there.

5. In algorithm 2, it could be helpful to explain the asymmetry between the initializations of parameters p and q. Why p is fixed as 1/2 and only q is initialized according to the weights?

6. In the discussion of  experimental results, it is suggested that the superiority of AMLE is because it takes into account the voters' reliabilities. Then, is it possible that a simpler weighted majority aggregation rule (with initial parameters as weights) is sufficient to improve the accuracy but saving the time cost of the alternating optimization procedure?

7. In the conclusion, it is said that when applying to single-instance problems, it suffices to fix the prior parameters once. In such case, how these parameters are chosen?  Randomly or also some initialization procedures like the one for reliability parameters?



**Q7 Justification For Your Score:**

The novelty of the proposed mechanism and its experimental performance.

**Q9 Complying With Reviewing Instructions:**

1: Yes.

---

### Official Review · Reviewer_QMsV · 2022-04-07

**Q2(1) Originality/Novelty:** 3
**Q2(2) Significance/Impact:** 3
**Q2(3) Correctness/Technical Quality:** 3
**Q2(6) Clarity Of Writing:** 2
**Q6 Overall Score:** 5
**Q8 Confidence In Your Score:** 3

**Q1 Summary And Contributions:**

The authors study multiwinner voting in a setting where ground truth exists (i.e., there is an objectively correct winning committee) and the size of the winning committee is unknown. The authors assume that the votes represent the ground truth subjected to a certain noise model and propose an algorithm (multiwinner voting rule) that discovers the most likely ground truth. To this end, they use an EM-like algorithm.

**Q2 Assessment Of The Paper:**

More detailed information regarding each of these aspects is given below:

**Q2(4) Quality Of Experiments (Optional):**

3: Good: The experimental evaluation is adequate, and the results convincingly support the main claims.

**Q2(5) Reproducibility:**

3: Good: Key resources (e.g., proofs, code, data) are available and key details (e.g., proofs, experimental setup) are sufficiently well-described for competent researchers to confidently reproduce the main results.

**Q3 Main Strengths:**

1) The authors show a multiwinner voting setting with a fairly realistic application
2) The results took some work to obtain (in principle, it seems that doing this work was standard, but it still was work and I am sure that there were a number of difficulties and obstacles that the authors faced, which are not really presented in the paper since the authors managed to overcome them).

**Q4 Main Weakness:**

1) The writing is of fairly low quality. In particular, one does not get a natural narrative flow where it is clear why certain things are done. For example, formulas from Def. 1 are unintuitive until a much later point in the paper.

2) The authors do not really comment on their experimental results and do not offer much conclusions.

**Q5 Detailed Comments To The Authors:**

Please proofread your paper carefully and comment in more details on Figures 2-4.

**Q7 Justification For Your Score:**

This paper does something useful, but does not analyze its own experiment very well.

**Q9 Complying With Reviewing Instructions:**

1: Yes.

---

### Official Review · Reviewer_jrbw · 2022-04-11

**Q2(1) Originality/Novelty:** 2
**Q2(2) Significance/Impact:** 2
**Q2(3) Correctness/Technical Quality:** 3
**Q2(6) Clarity Of Writing:** 3
**Q6 Overall Score:** 3
**Q8 Confidence In Your Score:** 4

**Q1 Summary And Contributions:**

The paper addresses the question of epistemic voting: recovering a ground truth when the voters are unbiased estimators, voting is based on an approval procedure, outputs multiple winners, and where the number of winners is subject to cardinality constraints. It proposes an algorithm similar to EM to estimate the ground truth. The performance of the proposed algorithm is assessed experimentally using data specifically obtained from human annotators, with somewhat satisfying results.

**Q10 Ethical Concerns (Optional):**

I have concern with the sentence "the ground truth consists of those patients who most deserve to be cured" found in the introduction, in the context of deciding who receives intensive medical care. I think this example of application can be completely removed from the paper without impacting negatively the contribution, and so it should.

**Q2 Assessment Of The Paper:**

More detailed information regarding each of these aspects is given below:

**Q2(4) Quality Of Experiments (Optional):**

2: Fair: The experimental evaluation is weak: important baselines are missing, or the results do not adequately support the main claims.

**Q2(5) Reproducibility:**

3: Good: Key resources (e.g., proofs, code, data) are available and key details (e.g., proofs, experimental setup) are sufficiently well-described for competent researchers to confidently reproduce the main results.

**Q3 Main Strengths:**

Except for the items mentioned in Q4 and Q10, the paper is well written and easy to follow. The research question is well motivated. The results seem correct.

**Q4 Main Weakness:**

The contribution is presented in an unsatisfying manner.
The estimation problem is cast in the MLE framework, with a latent ground truth and latent parameters governing the accuracy of voters and the prior probability of an alternative to be approved into the ground truth. Maximization is addressed with an alternating algorithm very similar to EM: fixing the parameters to estimate the ground truth / estimating the parameters knowing the ground truth. This is a fine algorithm, but the output is only a local maximum of the likelihood. Proposing soft computing strategies is fine, when addressing difficult optimization problems. What is not is claiming optimality in the abstract and not discussing this point in the introduction.

**Q5 Detailed Comments To The Authors:**

1. Writing unapologetically that "only local optimality is guaranteed, as classical in optimization" just shows that the writer is ignorant, disrespectful w.r.t. the domain of Optimization, and proud of it.
Imho, this could be easily fixed by mentioning beforehand that the maximization problem seems very difficult, thus warranting a heuristic approach. Of course, some evidence that the optimization problem is indeed difficult, e.g. because it is intractable and difficult to approximate, would give the contribution much more weight.

2. This approach looks very similar to EM to me. The authors claim it is indeed a simplification of EM that would be easier to explain to voters. This statement should appear in the introduction (or even in the abstract), so that an informed reader cognizant of EM would get a clear grasp of the contribution much more quickly. Also: in what sense is it simplifying?

**Q7 Justification For Your Score:**

The main weakness reported above requires a major revision.

**Q9 Complying With Reviewing Instructions:**

1: Yes.

---

### Official Review · Reviewer_B9Pw · 2022-04-13

**Q2(1) Originality/Novelty:** 3
**Q2(2) Significance/Impact:** 3
**Q2(3) Correctness/Technical Quality:** 3
**Q2(6) Clarity Of Writing:** 3
**Q6 Overall Score:** 6
**Q8 Confidence In Your Score:** 2

**Q1 Summary And Contributions:**

This paper presents an epistemic approach for multi-winner approval setting. The approach investigates a noise model which makes use of the prior beliefs based on ground truth. The paper then presents an explainable, EM based algorithm for estimating ground truth based on voter noise and prior belief.

**Q2 Assessment Of The Paper:**

More detailed information regarding each of these aspects is given below:

**Q2(4) Quality Of Experiments (Optional):**

3: Good: The experimental evaluation is adequate, and the results convincingly support the main claims.

**Q2(5) Reproducibility:**

3: Good: Key resources (e.g., proofs, code, data) are available and key details (e.g., proofs, experimental setup) are sufficiently well-described for competent researchers to confidently reproduce the main results.

**Q3 Main Strengths:**

+ the paper is based on sound theoretical motivation
+ the proposed approach works fairly well

**Q4 Main Weakness:**

- The paper contains a small scale empirical validation.
- Limited experimental comparisons to previous work



**Q5 Detailed Comments To The Authors:**

I believe, in general adding experimental comparison to previous work such as Caragiannis et al 2020 would be helpful to help make a compelling case.

**Q7 Justification For Your Score:**

The paper is based on sound fundamentals and the exposition is fairly good. The experimental results are encouraging.

**Q9 Complying With Reviewing Instructions:**

1: Yes.

---

### Decision · Program_Chairs · 2022-05-15

**Decision:**

Accept (Poster)

**Comment:**

Meta Review: This was a difficult paper to evaluate. The application was good. However, some of the description and claims were not as clear as they could have been (as acknowledged in the author's response).